# DEEP CURIOSITY SEARCH:
# INTRA-LIFE EXPLORATION
# CAN IMPROVE PERFORMANCE ON CHALLENGING
# DEEP REINFORCEMENT LEARNING PROBLEMS

## ABSTRACT

Traditional exploration methods in reinforcement learning (RL) require agents to perform random actions to find rewards. But these approaches struggle on sparse-reward domains like Montezuma's Revenge where the probability that any random action sequence leads to reward is extremely low. Recent algorithms have performed well on such tasks by encouraging agents to visit new states or perform new actions in relation to all prior training episodes (which we call across-training novelty). But such algorithms do not consider whether an agent exhibits intra-life novelty: doing something new within the current episode, regardless of whether those behaviors have been performed in previous episodes. We hypothesize that across-training novelty might discourage agents from revisiting initially non-rewarding states that could become important stepping stones later in training—a problem remedied by encouraging intra-life novelty. We introduce Curiosity Search for deep reinforcement learning, or Deep Curiosity Search (DeepCS), which encourages intra-life exploration by rewarding agents for visiting as many different states as possible within each episode, and show that DeepCS matches the performance of current state-of-the-art methods on Montezuma's Revenge. We further show that DeepCS improves exploration on Amidar, Freeway, Gravitar, and Tutankham (many of which are hard exploration games). Surprisingly, DeepCS also doubles A2C performance on Seaquest, a game we would not have expected to benefit from intra-life exploration because the arena is small and already easily navigated by naive exploration techniques. In one run, DeepCS achieves a maximum training score of 80,000 points on Seaquest—higher than any methods other than Ape-X. The strong performance of DeepCS on these sparse- and dense-reward tasks suggests that encouraging intra-life novelty is an interesting, new approach for improving performance in Deep RL and motivates further research into hybridizing across-training and intra-life exploration methods.

## 1 INTRODUCTION

Deep reinforcement learning (Deep RL) has achieved landmark success in learning how to play Atari games from raw pixel input (Mnih et al. (2015; 2016); Schaul et al. (2016); van Hasselt et al. (2016)). Many of these algorithms rely on naïve exploration techniques that employ random actions to find new rewards: either by selecting completely random actions at a fixed probability (e.g. the $\epsilon$-greedy methods of Q-learning), or by sampling stochastically from the current policy's action distribution, as in policy gradient methods (Mnih et al. (2016)). However, in *sparse-reward* games like Montezuma's Revenge, no reward can be obtained at all until agents perform long sequences of complex behaviors such as descending ladders, crossing empty rooms, and avoiding enemies. Because the probability that any random action sequence will produce these behaviors is extremely low, a very large number of samples is required to find them; and while it is now possible to obtain many Atari samples in reasonable wall-clock time (Conti et al. (2018); Horgan et al. (2018); Salimans et al. (2017)), this approach is inefficient and will not scale well to harder tasks. For example, even after 22.8 *billion* Atari frames, Ape-X Horgan et al. (2018) (a highly parallel, naïve-exploration algorithm) cannot match the performance achieved by 400 million frames of directed, count-based

| | DQN | A3C | PC | PCn | R | Ape-X | DeepCS |
| Frames | 200M | 200M | 400M | 500M | 200M | 22.8B | 640M |
|---|---|---|---|---|---|---|---|
| Amidar | 739 | 283 | 964 | $\sim$900 | 5131 | 8659 | 1404 |
| Freeway | 30 | 0 | 30 | 31 | 34 | 33 | 33 |
| Gravitar | 306 | 269 | 246 | 859 | 1419 | 1598 | 881 |
| MR | 0 | 53 | 3439 | 3705 | 154 | 2500 | 3500 |
| Tutankham | 186 | 156 | 132 | $\sim$190 | 241 | 272 | 256 |
| Alien | 3069 | 518 | 1945 | $\sim$1700 | 9491 | 40804 | 2705 |
| Kangaroo | 6740 | 94 | 5475 | $\sim$7900 | 14637 | 1416 | 2837 |
| Pitfall | - | -78 | -155 | 0 | 0 | 0 | -186 |
| Private Eye | 1788 | 206 | 246 | 15806 | 4234 | 864 | 1105 |
| Seaquest | 5286 | 2300 | 2274 | $\sim$2500 | 19,176 | 392,952 | 3443 |
| Venture | 380 | 23 | 0 | 1356 | 5 | 1813 | 12 |
| Wizard of Wor | 3393 | 17244 | 3657 | $\sim$2500 | 17862 | 46897 | 2134 |

Table 1: **DeepCS outperforms pseudo-counts (PC), Q-learning (DQN), and policy gradients (A3C) on five Atari games (shown at top of table) including Montezuma's Revenge (MR) and outperforms Ape-X and Rainbow (R) on MR.** Approximate results ($\sim$) were taken from training curves where no tabular results were available. DeepCS remains competitive with follow-up work on pseudo-counts (PCn). Median scores are shown here for DeepCS; the best-performing runs produced much higher scores including 6600 points on MR, 2600 points on Amidar, and 80,000 points on Seaquest (SI Fig. S4).

exploration (Bellemare et al. (2016)) on Montezuma's Revenge (Table 1, Ape-X vs. PC). Sparse-reward environments thus tend to require a prohibitively large number of training steps to solve without *directed* (i.e. intelligent) exploration (Bellemare et al. (2016); Kulkarni et al. (2016); Tang et al. (2017)).

Many approaches have tried to produce better exploration in RL (Bellemare et al. (2016); Brafman & Tennenholtz (2001); Conti et al. (2018); Houthooft et al. (2018); Kearns & Singh (2002); Kulkarni et al. (2016); Oudeyer et al. (2007); Schmidhuber (2010); Tang et al. (2017)). These methods typically encourage agents to visit novel states or perform novel actions, requiring the algorithm to record how often states have been visited (e.g. by maintaining a *count*, which is optimal in tabular RL (Kolter & Ng (2009)), but requires further abstractions to scale to high-dimensional state spaces (Bellemare et al. (2016); Tang et al. (2017))). Notably, these techniques record state visitations over the entirety of training time, requiring agents to express novelty in relation to all prior training episodes (which we call *across-training novelty*). For example, in an algorithm based on pseudo-counts (Bellemare et al. (2016)), the reward for visiting the same state in a future game is almost always lower than in the previous episode, encouraging agents to seek out states that have not been encountered in prior training iterations. Similarly, intrinsic motivation approaches that reward agents e.g. for visiting poorly-compressed states (Schmidhuber (2010)) tend to compress the same state better each time it is seen, meaning that future visits to that state may not be rewarded much (or at all) if the state has been visited in prior episodes. This observation also applies to the application of Novelty Search to Deep RL (Conti et al. (2018); Lehman & Stanley (2011)), which calculates the novelty of a behavior by comparing it against an archive of prior agents (Lehman & Stanley (2011)); agents must perform new actions in relation to all their ancestors.

Across-training approaches do not examine novelty within each *lifetime* (called *intra-life* novelty (Stanton & Clune (2016))). In our Atari experiments, we define a lifetime as the complete rollout for each game (which can span multiple in-game lives), although we note that this definition is a hyperparameter choice. Making behaviors uninteresting solely because they have been performed in prior lifetimes can sometimes be detrimental: for example, it is beneficial for each human to learn their native language, even though many prior humans have already mastered that task. This may be especially important in multi-task environments (Conti et al. (2018); Stanton & Clune (2016)). Consider a cross-shaped maze where to go in each cardinal direction, agents must master an entirely different skill (swimming to go west, climbing mountains to go east, etc.) and *all* skills must be acquired to solve a larger task (Stanton & Clune (2016)). With across-training novelty, if an individual starts by exploring the west branch, it may continue to do so (learning how to swim) until state

visitations become too high, and then explore a less-visited branch (learning how to climb mountains) where the exploration bonus is higher (Conti et al. (2018); Stanton & Clune (2016)). Because visitations never decrease, the west branch (and swimming behavior) might never be revisited. The algorithm may explore the entire environment, but due to catastrophic forgetting (French (1999)) the final agent is likely a *specialist* (Stanton & Clune (2016)); it can perform only the most recent skill it learned and cannot solve the larger task requiring *all* skills.

Even if catastrophic forgetting can be eliminated, across-training novelty may not make progress if the required skills are hierarchical, or if a state visited too early (offering no immediate reward) is later critical for solving a task. For example, suppose an agent must open a door at the end of a hallway, but first requires a key located elsewhere in the environment. If the agent spends much too time initially exploring the hallway and only later discovers the key, it may not have any incentive to return to the hallway and the door-opening behavior might never be learned (assuming state conflation i.e. that the hallway is still considered explored even after the key is obtained, which we expect would occur with deep neural network function approximators as it does with animals). Designing reward functions or state representations to avoid such pitfalls can be difficult (Ng et al. (1999); Sutton & Barto (1998)), especially on any complex (often deceptive) task (Nguyen et al. (2015); Woolley & Stanley (2011)). Experience replay does not alleviate this problem because it only helps once rewards have been discovered; but we may need directed exploration to discover such rewards in the first place. While state counts could be decayed over time to refresh "interest," choosing which states should be interesting again (and when) is a nontrivial, domain-specific task.

An intra-life algorithm might avoid these issues because it *conflates* different exploration strategies. For example, an agent that first explores the hallway and then finds the key could receive identical intrinsic rewards as an agent that explores these states in reverse. Because both behaviors are similarly encouraged, it is more likely that learning will periodically switch between them and ultimately discover the greater value of obtaining the key first. Additionally, population-based search algorithms would not prefer one trajectory over the other; an intra-life algorithm could thus preserve a diversity of different behaviors (Stanton & Clune (2016)), making it easier to find the correct ordering since neither behavior is "locked out" by the history of exploration in prior episodes.

We investigate the possibility of directly rewarding *intra-life* novelty: encouraging agents to explore as many states as possible within their lifetime. Using the previous example, even if the hallway is explored over many different games, an intra-life algorithm resets the exploration bonus such that the hallway becomes "interesting" again at the start of every new game. This is the approach taken by Curiosity Search (Stanton & Clune (2016)), which promotes intra-life novelty by borrowing from biological principles of intrinsic motivation (Berlyne (1960); Dayan & Belleine (2002); Kakade & Dayan (2000)). In vertebrates, this motivation is generally characterized by a dopamine response for behaviors that produce unexpected results (i.e. a high temporal difference error) (Kakade & Dayan (2000)), encouraging the discovery of novel states and creating an instinct to play/explore that may facilitate an animal's survival by leading to the acquisition of useful skills (Benson-Amram & Holekamp (2012); Berlyne (1960); Kohler (1924); Špinka et al. (2001)). Curiosity Search expresses this idea by first quantifying the types of rewarded behaviors and secondly rewarding agents for expressing as many of those behaviors as possible within each lifetime (Stanton & Clune (2016)). An *intra-life novelty compass* can optionally be provided to direct agents towards novel behaviors and speed up learning, but is not required for the algorithm to function (Stanton & Clune (2016)). Curiosity Search produced better overall performance than an across-training exploration algorithm (Novelty Search (Lehman & Stanley (2011))) and improved the acquisition of door-opening skills in a maze environment (Stanton & Clune (2016)). However, Curiosity Search was only investigated on a simple task and trained very small networks ($\sim$100 parameters), in contrast to the millions of parameters that must be learned in Deep RL.

In this work, we adapt Curiosity Search to high-dimensional, challenging RL games. Specifically, we discretize the pixel space of Atari games such as Montezuma's Revenge into a *curiosity grid* and reward agents for visiting new locations on this grid. The grid is periodically reset such that every location becomes "interesting" again at the start of each game, thereby encouraging agents to express high intra-life novelty. We show that this adaptation, called DeepCS, matches the average performance of state-of-the-art methods on the hard-exploration game Montezuma's Revenge and also performs well on Amidar, Freeway, Gravitar, Tutankham, and Seaquest (in the latter of which DeepCS doubles A2C performance and achieves up to 80,000 points in one run). The strong performance of

DeepCS on these sparse- and dense-reward tasks suggests that encouraging intra-life novelty is an interesting new approach for producing efficient domain exploration, and opens up new directions towards hybridizing across-training and intra-life exploration strategies.

## 2 METHODS

We primarily examine performance of DeepCS on two Atari games: Montezuma's Revenge and Seaquest. The first is a notoriously difficult sparse-reward game in which exploration is critical to success. Naïve exploration strategies have trouble solving Montezuma's Revenge due to the low frequency of rewards and the long sequences of complex behaviors required to obtain them (Bellemare et al. (2016); Kulkarni et al. (2016); Tang et al. (2017)), making it an ideal candidate for demonstrating the benefits of intra-life exploration. Seaquest instead has a dense-reward space easily learned via traditional approaches (Mnih et al. (2015; 2016); van Hasselt et al. (2016)), allowing us to examine the effects of intra-life exploration on games in which such techniques are not seemingly required.

We also examine performance on 10 additional games: Alien, Amidar, Freeway, Gravitar, Kangaroo, Private Eye, Pitfall, Tutankham, Venture, and Wizard of Wor. Though not as infamously difficult as Montezuma's Revenge, many of these games are hard-exploration tasks according to the taxonomy in (Ostrovski et al. (2017)) and are interesting challenges for intra-life exploration. We primarily focus our discussion on Montezuma's Revenge because the latter has become a notoriously difficult challenge in RL; only a handful of techniques (Bellemare et al. (2016); Horgan et al. (2018); Kulkarni et al. (2016); Ostrovski et al. (2017); Tang et al. (2017)) are able to reliably exit the first room (achieving 400 points), in contrast to human players who explore many rooms and achieve a mean score of 4367 points (Mnih et al. (2015)).

In this work, we translate the general idea of Curiosity Search ("do something new" (Stanton & Clune (2016))) into a reward for agents that "*go* somewhere new" in the game world. To do so, we divide the game world of Atari into a *curiosity grid*, each tile of which provides a reward when visited by the agent (Fig. 1). We then obtain the agent's position from game RAM to indicate where on the grid an agent has visited; if the game has multiple rooms, as in Montezuma's Revenge, a distinct grid is created for each room. This approach is similar to the bitmasking of states performed in (Kulkarni et al. (2016)) with the main exception that we did *not* manually specify any objects of interest (e.g. the key in Montezuma's Revenge); the grid is constructed from equally sized square tiles given an arbitrary *grid size*, discussed below. While we would ultimately prefer that agents learn to remember where they have been automatically from pixel input, extracting this information from RAM is simpler and allows us to directly examine whether encouraging intra-life novelty can help solve difficult exploration problems in RL. In future work, we could encourage the agent to learn directly from pixels where it has been (or, more generally, what it has done) e.g. with recurrent neural networks trained with an auxiliary task like outputting the curiosity grid per time-step. If one did not have access to RAM states, position could be inferred via object detection (Fragkiadaki et al. (2015)) or by implicit state-space representations such as Skip Context Tree Switching (Bellemare et al. (2014)), the latter of which was shown to work well in a roughly similar counting-based model (Bellemare et al. (2016)). Also interesting is investigating under what conditions such memories of past behaviors might emerge on their own because they are helpful for exploration (e.g. in meta-learning (Duan et al. (2018))).

Agents were trained via A2C (Mnih et al. (2016)) using the default hyperparameters in OpenAI Baselines (Dhariwal et al. (2017)). The clipped game reward at any state $R_s$ was replaced with $\hat{R}_s = \beta R_s + (1 - \beta)I_s$, where $I_s$ is the intrinsic reward (1 if an unexplored tile was just touched and 0 otherwise), $R_s$ is the regular game reward clipped to $[-1, 1]$ (Mnih et al. (2015)), and $\beta$ is a constant in the range $[0, 1]$ that controls the relative weight of intrinsic and regular game rewards. In the untuned case (where both rewards are equal) $\hat{R}_s = clip(R_s + I_s)$. We chose $\beta = 0.25$ after a coarse hyperparameter sweep in Montezuma's Revenge and used untuned values in all other games. Although we treat exploration as binary in this work, $I_s$ could also be reduced gradually in proportion to how many times a tile has been visited, similar to the mechanisms in other non-binary count-based approaches (Bellemare et al. (2016); Brafman & Tennenholtz (2001); Kearns & Singh (2002); Ostrovski et al. (2017)).

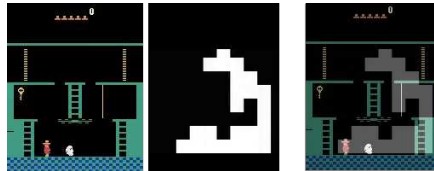

Figure 1: **DeepCS encourages agents to visit new places in Montezuma's Revenge.** White sections in the *curiosity grid* (middle) show which locations have been visited; the unvisited black sections yield an exploration bonus when touched. The network receives both game input (left) and curiosity grid (middle) and must learn how to form a map of where the agent has been (hypothetical illustration, right). The grid is reset when the agent loses all lives and starts a new game, encouraging *intra-life* exploration irrespective of previous games.

The intra-life novelty compass of Curiosity Search is traditionally a vector indicating the direction of the nearest rewards (Stanton & Clune (2016)). Because direction is more appropriate for the first person view of a driving robot (Lehman & Stanley (2011); Stanton & Clune (2016)) and less so for Atari games (in which the agent sees the entire "world" at once), we instead use the curiosity grid as an intra-life novelty compass, providing it as an extra network input. The agent must learn how to use this "map" to direct itself towards unexplored areas of the game. The curiosity grid is constructed using a per-tile grid size of 16x16 for most games, as described in SI Sec. S1. To ensure dimensionality equal to normal game state inputs, the grid is resized to match the 84x84 downsampled game screen (Mnih et al. (2015)). As discussed below, we found that this extra input was helpful, but not necessary to benefit DeepCS on hard RL tasks; agents perform well if rewarded for visiting new areas even when they do not see where they have (or have not) already been (Fig. 5).

All algorithms were trained for 160 million training steps with a frame skip of 4 (Mnih et al. (2015)), resulting in 640 million game frames per run. 25 runs were performed for each experiment. Plotted results show medians (solid lines), interquartile range (shaded areas), and minimum/maximum scores (dotted lines). We compare DeepCS (with an underlying A2C implementation) against A2C alone; the latter is a synchronous and equally-performing implementation of A3C (Mnih et al. (2016)). Our A2C algorithm uses 16 actors exploring in parallel; training curves were obtained by examining the most recent episode score of each actor and plotting the averaged result. Statistical comparisons were obtained via a two-tailed Mann-Whitney U test, which does not assume normality. Videos of end-of-run agents, open source code, and plot data can be obtained from: keptanonymousforreview.

As with any exploration method, encouraging intra-life novelty has possible disadvantages. In a maze environment, Curiosity Search will reward agents for exploring every dead end, even though such states are likely useless for solving the maze itself. Our hypothesis is not that the intra-life exploration of Curiosity Search is superior to that of across-training novelty, but rather that the former is a valuable and interesting tool for solving challenging exploration problems in RL. We also note that while DeepCS might encourage inefficient exploration, (1) this may not prove problematic as the algorithm still values real game rewards (how much is a hyperparameter choice) and may find higher ultimate value in ignoring some dead ends, and (2) one could use the exploration produced by DeepCS to later produce a distilled, exploitive, efficient policy (e.g. by annealing the weight on intrinsic rewards at the end of training or filling an experience replay buffer with states observed via DeepCS agents and then training a DQN agent off-policy to only maximize real game rewards).

## 3 RESULTS & DISCUSSION

In Montezuma's Revenge, DeepCS achieved a median game score of 3500 points (Fig. 2, top row), matching the performance of current state-of-the-art approaches (Table 1, DeepCS vs. PC (Bellemare et al. (2016)) and PCn (Ostrovski et al. (2017))) and significantly outperforming the naïve exploration strategy of A2C alone ($p < 0.00001$). The A2C control found the key in a handful of episodes (as occurs in A3C (Mnih et al. (2016))), but this event was too rare to be visible in the aggregate performance of 25 runs. We recorded intrinsic rewards for all treatments, but the A2C control did *not* receive those rewards during training. In further examination, we discovered that swapping the A2C implementation with a different algorithm (ACKTR (Wu et al. (2017))) allowed DeepCS to

achieve 6600 points in one run (tying the highest-reported single-run score (Bellemare et al. (2016))), although ACKTR was less stable overall than DeepCS+A2C (SI Fig. S2).

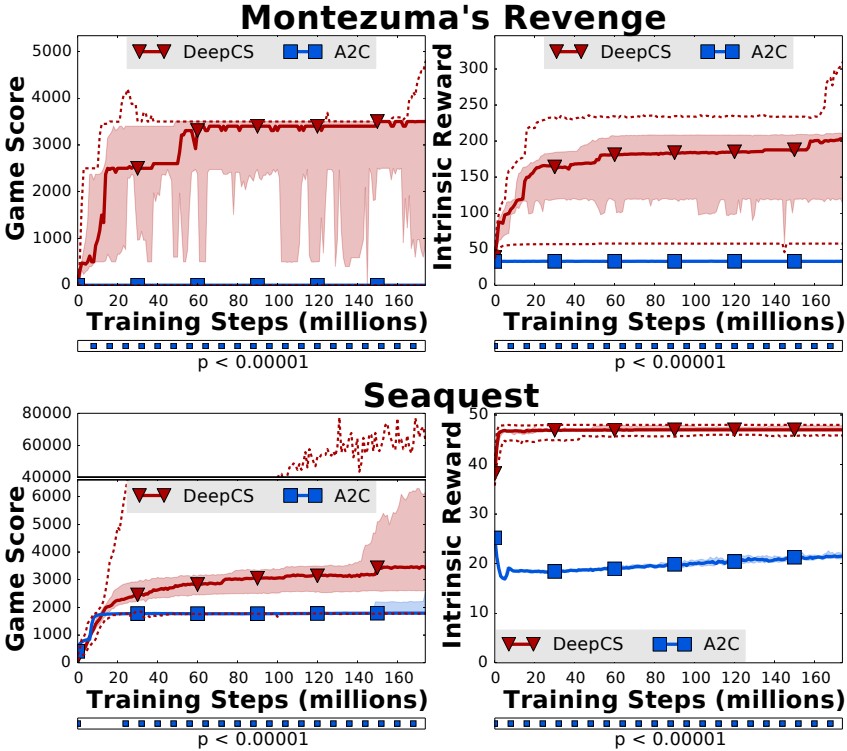

Figure 2: **DeepCS improves performance on sparse- and dense-reward Atari games.** In Montezuma's Revenge (a challenging sparse-reward game), DeepCS vastly outperforms the naïve exploration of A2C alone and matches the average performance of state-of-the-art methods (Table 1, DeepCS vs. PC and PCn). On Seaquest (an easier, dense-reward game), DeepCS nearly doubles the median performance of A2C (3443 vs. 1791 points) and obtains nearly 80,000 points in one run. The rightmost plots show intrinsic rewards, quantifying how much of the game world has been explored; horizontal bars below each plot indicate statistical significance. For additional games, see SI Fig. S1.

Barring an extremely large amount of training data (22 billion game frames for Ape-X, Table 1), DQN and many other algorithms receive almost no reward on this game (Conti et al. (2018); Mnih et al. (2015; 2016); Salimans et al. (2017); Schaul et al. (2016); van Hasselt et al. (2016)), although A3C Mnih et al. (2016) can occasionally find the key and Rainbow (Hessel et al. (2017)) infrequently exits the first room. Even promising methods that improve exploration in Deep RL still struggle on this task (Kulkarni et al. (2016); Tang et al. (2017)), sometimes also yielding a score of zero (Conti et al. (2018)); Montezuma's Revenge has resisted many concerted efforts to solve it, remaining a very difficult game even for intelligent exploration techniques. The strong performance of DeepCS on this task suggests that encouraging intra-life exploration is an exciting new means of producing better performance in sparse-reward domains.

The intrinsic rewards obtained by DeepCS were significantly higher than those of A2C alone, indicating that DeepCS agents explore more of the game world (Fig. 2, top-right, $p < 0.00001$). These rewards rose slightly at the end of training, but we did not possess the computational resources to examine whether this small increase in exploration would continue and possibly translate into even higher game scores. We note that while the DeepCS game score has a wide interquartile range, intrinsic rewards were far more stable (Fig. 2, top-left vs. top-right). This likely occurs because A2C policies are stochastic; a small variation in the action output might cause an agent to narrowly "miss" rewards during evaluation (e.g. by missing one jump to collect an item, an agent might lose out on 1000 points), even when the policy is otherwise exploring well. When evaluating end-of-run agents, we discovered that this variance also produced much *higher* game scores (e.g. in one run, performance increased from 3500 to 5000 points depending on the starting random seed, SI Fig. S4).

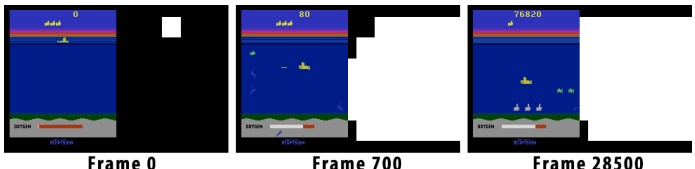

Figure 3: **DeepCS produces agents that explore a large percentage of Montezuma's Revenge.** The game starts in the upper-center room; filled sections indicate explored rooms and blank areas are unexplored. The best DeepCS agent explores 15 rooms (right), matching state-of-the-art techniques (Bellemare et al. (2016)). Most algorithms barely explore 1-2 rooms of this very difficult game. Agents that move west from the initial room (left) seem rare in DeepCS (3 out of 25 runs) and in the literature.

Figure 4: **We did not expect DeepCS to help Seaquest because the curiosity grid can quickly saturate.** At game start, the grid is unfilled and many intrinsic rewards can be obtained (left). However, after only 700 game frames (middle), the grid is nearly saturated; DeepCS can only provide training feedback for a few hard-to-reach tiles. However, even this brief presence of intrinsic rewards allows agents to learn behaviors affording them e.g. 76,000 points on Seaquest in a single run (right).

Agents produced by DeepCS explore a large percentage of the first level in Montezuma's Revenge, which contains 24 rooms in a pyramid shape. The best-performing DeepCS agent explored a total of 15 rooms (Fig. 3, right), matching the best-agent exploration of state-of-the-art methods (Bellemare et al. (2016); Ostrovski et al. (2017)). In contrast, most traditional techniques only rarely exit the first room (Conti et al. (2018); Mnih et al. (2015; 2016); Salimans et al. (2017); Schaul et al. (2016); Tang et al. (2017); van Hasselt et al. (2016)) and at most are able to explore two (Kulkarni et al. (2016)) (except for Ape-X (Horgan et al. (2018))). We also observed 3 runs in which agents moved west out of the initial room (Fig. 3, left), which seems to be a rare occurrence on this game (we are not aware of any papers/videos demonstrating west-moving behavior except for (Kulkarni et al. (2016))).

While DeepCS improves performance in the sparse-reward environment of Montezuma's Revenge, it is unclear what effects intra-life exploration might have on a dense-reward game like Seaquest. One might expect DeepCS's intrinsic rewards to be ignored as background noise; exploring distant game locations does not directly lend itself to the task of shooting submarines, rescuing divers, or surfacing for air. Furthermore, the arena of Seaquest contains only one small "room" and the curiosity grid can quickly saturate, causing intrinsic rewards to drop towards zero (Fig. 4). Because fewer locations can be visited (and rewarded by DeepCS) in Seaquest than in multi-room games such as Montezuma's Revenge, and because Seaquest's regular game rewards are easy to obtain, one might expect that DeepCS's training gradients would be too infrequent or small to affect exploration/performance.

Surprisingly, we found that DeepCS almost doubled the final median A2C performance on Seaquest (3443 vs. 1791 points, Fig. 2 bottom row, $p < 0.00001$). In one run, DeepCS obtained almost 80,000 points, a score which to our knowledge is superseded only by Ape-X (which requires collecting far more samples, Table 1). Performance did not plateau in this run even after 160 million training steps; we also found that depending on random start conditions, the final agent could obtain up to 278,130 points during evaluation (SI Fig. S4). DeepCS's upper quartile score also began to improve rapidly after 140 million training iterations, suggesting that median performance might continue to rise given more training time (and may possibly match state-of-the-art results if parallelized like Ape-X). One possible explanation for DeepCS's success on Seaquest is that encouraging agents to visit all game locations improves their ability to navigate the game world, which may then enable the discovery of better game behaviors (e.g. dodging and surfacing for air) than are normally found by the greedy focus of obtaining game rewards only. However, further work is required to verify this hypothesis.

One might argue that providing the curiosity grid as input is "cheating" because the agent has access to a perfect memory of where it has been. Other methods use novelty measures solely to determine rewards without changing the original Atari pixel input (Bellemare et al. (2016); Ostrovski et al.

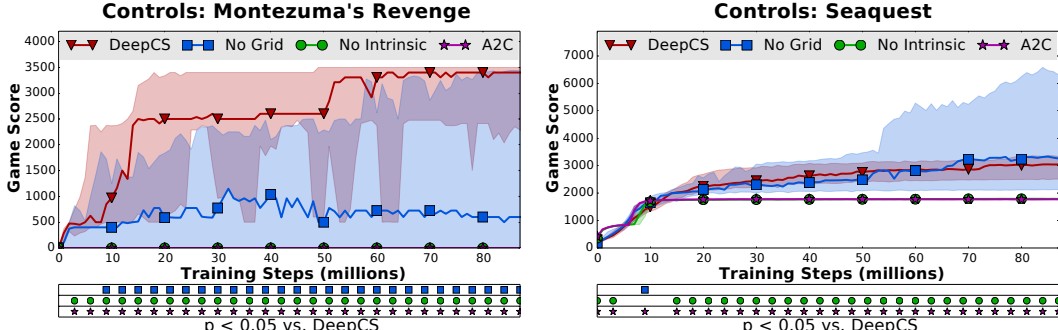

Figure 5: **DeepCS does not need the curiosity grid input to improve domain exploration.** When the curiosity grid has been removed (No Grid), agents perform equally well on Seaquest; performance drops on MR but remains better than A2C alone. Both games suffer when intrinsic rewards are removed (No Intrinsic), suggesting that intrinsic rewards (not the grid) are the key aspect of DeepCS.

(2017); Tang et al. (2017)). The original work on Curiosity Search (Stanton & Clune (2016)) also hypothesized that an intra-life novelty compass is not always required. We investigate both claims by examining performance of DeepCS (1) with only the curiosity grid (no intrinsic rewards) and (2) with only intrinsic rewards (no curiosity grid). If the intra-life novelty compass is not essential, we would expect DeepCS to perform just as well after the grid is removed. It is also possible that the grid and intrinsic rewards must work together for DeepCS to succeed; we can determine whether this combination is required by removing only intrinsic rewards.

Removing intrinsic rewards from DeepCS significantly hurt performance on both Montezuma's Revenge and Seaquest ($p < 0.05$, Fig. 5, No Intrinsic). However, removing the grid while keeping intrinsic rewards had almost no effect on Seaquest performance at all ($p \geq 0.05$, Fig. 5, No Grid) and performance remained higher than A2C alone. Removing the grid from Montezuma's Revenge significantly reduced performance ($p < 0.05$), but we observed a very wide interquartile range; at least 25% of No Grid agents were able to match the performance of DeepCS with a grid. Further examination showed that removing the grid prevented some runs from solving the first room; but those that *did* exit the first room continued on to match DeepCS performance (SI Fig. S3). While the curiosity grid can be very useful on hard exploration tasks containing sparse rewards, we conclude that such a grid is not necessary; DeepCS still improves domain exploration without it.

DeepCS improved performance when compared against other directed-exploration techniques on Amidar, Freeway, Gravitar, and Tutankham (SI Fig. S1). The intra-life exploration of DeepCS did not perform nearly as well on the other games, but still did better than naïve exploration for most games (e.g. DeepCS vs. A3C, Table 1). Due to limited computational resources, we leave a more-detailed investigation of these and other Atari games to future work.

## 4 CONCLUSIONS & FUTURE WORK

Encouraging efficient domain exploration is a difficult problem in RL. Directed exploration strategies (Bellemare et al. (2016); Brafman & Tennenholtz (2001); Conti et al. (2018); Kearns & Singh (2002); Kulkarni et al. (2016); Oudeyer et al. (2007); Schmidhuber (2010); Tang et al. (2017)) often try to remedy this problem by encouraging novelty in the state- or action-space over all of training (which we call *across-training* novelty). These algorithms do not consider whether agents express *intra-life* novelty, doing something new within the context of the current episode. We hypothesized that across-training approaches might prematurely "lose interest" in later-valuable, but not immediately useful, states; a problem remedied by intra-life novelty because the latter encourages agents to "do everything once" in each lifetime (episode).

In this work, we introduced an algorithm to encourage intra-life exploration (DeepCS) by adapting Curiosity Search to Deep RL. On Montezuma's Revenge, in which traditional naïve-exploration algorithms receive no reward (Conti et al. (2018); Mnih et al. (2015; 2016); Salimans et al. (2017); Schaul et al. (2016); van Hasselt et al. (2016)) without billions of game samples Horgan et al. (2018), DeepCS matched performance of state-of-the-art, across-training exploration methods (Bellemare

et al. (2016); Ostrovski et al. (2017)) and outperformed those algorithms on Amidar, Freeway, Gravitar, and Tutankham. DeepCS doubled A2C performance on Seaquest and produced one agent achieving nearly 80,000 points, suggesting that intra-life exploration can improve policies on easier games that we might not initially expect to benefit from it. Finally, we showed that a *curiosity grid* (as agent memory) can improve exploration; this component might also improve state-of-the-art methods like pseudo-counts (Bellemare et al. (2016); Ostrovski et al. (2017)).

In future work, we propose replacing the RAM-determined agent positions used in this paper with more natural representations e.g. an autoencoder "hash" (Tang et al. (2017)). Because the combination of across-training and intra-life novelty produced better results than intra-life novelty alone in a simple maze environment (Stanton & Clune (2016)), combining both forms of novelty might offer similar benefits in Deep RL. Mixing rewards with a Monte Carlo return can potentially improve exploration (Bellemare et al. (2016)); adding this return to DeepCS may provide similar benefits. Because DeepCS might encourage exploration at the *expense* of obtaining rewards in sparse domains, we also wish to investigate combining DeepCS with a more exploitative algorithm—the former directing exploration and the latter using explored states (e.g. as experience replay) to learn a policy dedicated to obtaining game rewards.

Another interesting question is whether other forms of novelty (e.g. in input/output pairs (Gomes & Christensen (2013))) can improve exploration. However, pressuring agents to reach new locations (as in this work) may itself be a good prior when we do not know which behaviors to reward or how to quantify them (Stanton & Clune (2016)). For example, in a robotics task it can be difficult to quantify different types of gaits such that agents learn many movement behaviors (sil), which may be needed if a servo becomes damaged (Cully et al. (2015)). However, requiring an agent to traverse different terrains (e.g. sand and concrete) in order to visit new areas may indirectly force the acquisition of different gaits because they are necessary for continued exploration.

We have shown that encouraging intra-life novelty is an interesting new tool for solving both sparse- and dense-reward deep RL problems. However, intra-life novelty is not necessarily superior to across-training approaches; each form of exploration has its own advantages and disadvantages. Ultimately, we believe that the combination of both ideas will be an important part of creating artificial agents that can better explore and improve performance on challenging deep RL domains.

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

## Supporting Information

### S1 Additional Experiment Details

The RAM offsets used for constructing the curiosity grid in each game are provided below, along with the resulting total grid size (prior to downsampling) and the per-tile size. The curiosity grid is then resized to 84x84 to match downsampled game screen input (Mnih et al. (2015)).

|  | X | Y | Room/Level | Grid Size | Per-Tile Size | Reward Weight |
|---|---|---|---|---|---|---|
| Alien | 0xAD | 0xB4 | 0x80 | 73x120 | 16 | equal |
| Amidar | 0xC2 | 0xBB | see below | 125x147 | 16 | equal |
| Freeway | - | 0x8E | 0xE7 | 1x172 | 20 | equal |
| Gravitar | 0x9F | 0xA0 | see below | 155x157 | 32 | equal |
| Kangaroo | 0x91 | 0x90 | 0xA4 | 121x23 | 3 | equal |
| MR | 0xAA | 0xAB | 0x83 | 153x122 | 16 | $\beta = 0.25$ |
| Pitfall | 0xE1 | 0xE9 | see below | 141x224 | 16 | equal |
| Private Eye | 0xBF | 0xE1 | 0xBE | 128x100 | 16 | equal |
| Seaquest | 0xC6 | 0xE1 | - | 114x96 | 16 | equal |
| Tutankham | 0x87 | 0xC1 | see below | see below | 20 | equal |
| Venture | 0xD5 | 0x9A | 0xBE | 160x79 | 16 | equal |
| Wizard of Wor | 0xB7 | 0xAF | 0x84 | 121x101 | 16 | equal |

With the exception of minor adjustments in Montezuma's Revenge, Gravitar, Kangaroo, and Tutankham, a single set of hyperparameters was used for all games. Changes to the per-tile size were only made in games for which the default value (16) was too large/small for effective learning. For example, the height of each level in Kangaroo is only 23; a per-tile size of 16 would conflate nearly the entire vertical space, when in fact the agent needs to move upwards in order to solve the game.

In Amidar, an additional value (at offset 0xD7) determines a numerical offset which must be added to 0xC2 and 0xBB to obtain the agent's position address. There is no single RAM value for determining the current level; instead, we noted that the value at RAM offset 0xB6 oscillated every time a new level was encountered and used this to determine when to reset the curiosity grid.

Room information in Gravitar appears to be encoded in two different offsets (0x81 and 0x82), the first indicating the agent "mode" (top-down, side-scrolling, etc.) and the second indicating the current planet number.

Pitfall encodes room information via a 2-dimensional "code" governed by offsets 0xE1 and 0xE9.

In Tutankham, the value at 0xFB changes to 35 whenever a level is completed. The entire level (of size 145x256) is visible in memory at all times. To ensure that the agent did not gain an unfair advantage by seeing the entire level at once, we sliced this grid into a 145x27 region corresponding to the pixels currently visible on the game screen at each time step.

Freeway does not have an x-coordinate in RAM as the agent can only move up/down. Seaquest does not have a room address because the entire game takes place on a single screen.

### S2 Additional Atari Games

We also examined DeepCS on Alien, Amidar, Freeway, Gravitar, Kangaroo, Private Eye, Pitfall, Tutankham, and Venture. Ten runs were performed for each game. All games used equal (untuned) weighting between intrinsic and regular rewards.

DeepCS performed very well on Amidar, Freeway, Gravitar, and Tutankham, slightly outperforming other state-of-the-art exploration methods (Table 1, PC and PCn), although it did not perform nearly as well on the other games. Taken in combination with its improvements in Montezuma's Revenge and Seaquest (Fig. 2), we conclude that encouraging intra-life novelty can indeed be a useful tool for improving performance on hard exploration tasks. However, more research is required before DeepCS can be generally applied to the entire Atari benchmark.

## S3   DEEPCS WITH ACKTR

We also examined DeepCS with a different underlying implementation: ACKTR (Wu et al. (2017)), a similar policy-gradient method that generally has better sample-efficiency than A2C and is known to outperform A2C on some Atari tasks. Both algorithms produce *stochastic* policies, as opposed to Q-learning methods (Mnih et al. (2015)) in which action-selection is deterministic. We chose 16 actors for ACKTR; otherwise, hyperparameters were identical to the defaults provided by OpenAI Baselines (Dhariwal et al. (2017)).

While ACKTR appeared less stable both in game score and intrinsic reward, we found almost no pointwise statistical difference between ACKTR and A2C (SI Fig. S2, $p \geq 0.05$) Notably, within 10 million training steps ACKTR produced one agent that consistently achieved 6600 points for the rest of training and during final evaluation. This matches the maximum end-of-run score obtained in Bellemare et al. (2016) exactly; however, Bellemare et al. (2016) only obtained this maximum during end-of-run evaluations and *not* during training. While this maximum represents only one run, we know of no other approach in the current literature that produces such a high-performing agent on Montezuma's Revenge so early in training.

## S4   IMPORTANCE OF THE CURIOSITY GRID IN MONTEZUMA'S REVENGE

When the curiosity grid was removed from agent input, DeepCS performed significantly worse on Montezuma's Revenge (Fig. 5, left). To further investigate the difference between these treatments, we trimmed the dataset such that runs that *never* leave the first room (scoring $< 400$ points over all of training) are removed, allowing us to examine only those agents that *do* leave the first room.

When we only examine runs in which agents exit the first room, the statistical difference between DeepCS and DeepCS (No Grid) disappears after 70 million training steps (SI Fig. S3, $p \geq 0.05$). In other words, removing the curiosity grid caused fewer runs to exit the first room, but when any run *does* exit the first room, the No Grid treatment can perform just as well as DeepCS with the curiosity grid input. We hypothesize that the curiosity grid may be helpful because it provides a visual correlation between intrinsic reward signals and the agent's position on the grid—agents can easily learn to "follow the grid" to reach new areas. Learning to explore from intrinsic feedback alone (i.e. without the grid) is harder, but not impossible, reinforcing the hypothesis (Stanton & Clune (2016)) that an intra-life novelty compass is useful, but not essential, to Curiosity Search.

## S5   RESETTING THE CURIOSITY GRID

We examined the prospect of never resetting the curiosity grid (to simulate a count-based, across-training novelty approach). However, this modified algorithm obtained no rewards in Montezuma's Revenge. We believe that by never resetting the grid, intrinsic rewards "fall off" too quickly to be learned; and without intrinsic rewards, exploration converges to $\epsilon$-greedy.

We further examined a variant of DeepCS in which the grid was reset on each life loss instead of each completed game; this approach also obtained no rewards. Videos of the produced agents suggest that resetting the curiosity grid too frequently creates a local optimum; many "suicidal leaps" yield intrinsic reward, but they cause life loss and reset the grid, thereby creating a behavior loop.

## S6   BEST-POLICY DISTRIBUTIONS

Both A2C and ACKTR produce stochastic policies; the number of starting no-ops and the random seed itself can influence the sequence of actions performed by an agent. To better visualize expected game performance, we take the best DeepCS agent on every game, evaluate each of these agents with 100 different random starting seeds, and plot the resulting distribution.

In different start conditions, the best DeepCS/A2C agent on Montezuma's Revenge obtained up to 5000 points (above the 3500 training median, Fig. 2), although in one unlucky instance it obtained only 400 points (SI Fig. S4). The best ACKTR agent instead *always* received the same score—6600 points—regardless of random starting seed. We found that intrinsic rewards for this agent varied widely between 53 and 149 (not shown), suggesting a robust policy that can reliably obtain the same game score even when the total amount of exploration changes due to different random initializations.

More surprisingly, the best Seaquest agent (which obtained approximately 80,000 points by end-of-training, Fig. 2) was able to achieve scores as high as 278,130 points depending on the random starting conditions; and on Gravitar, 10% of the different random seeds resulted in $\geq 2000$ points, exceeding current state-of-the-art methods (Table 1, Ape-X). While these results are very impressive, it is unwise to rely on any single run to gauge performance and we stress that the overall median performance of DeepCS was not nearly so high (Fig. 2 and SI Fig. S1). On the other hand, had we chosen a sample size of 10 final evaluations e.g. on Gravitar, we might have come to the (incorrect) conclusion that our best agent always achieves state-of-the-art performance of $\geq 2000$ points. These plots reinforce the importance of examining the distribution of stochastic policies (with a sufficiently large sample size) rather than relying on any single point estimate of performance.

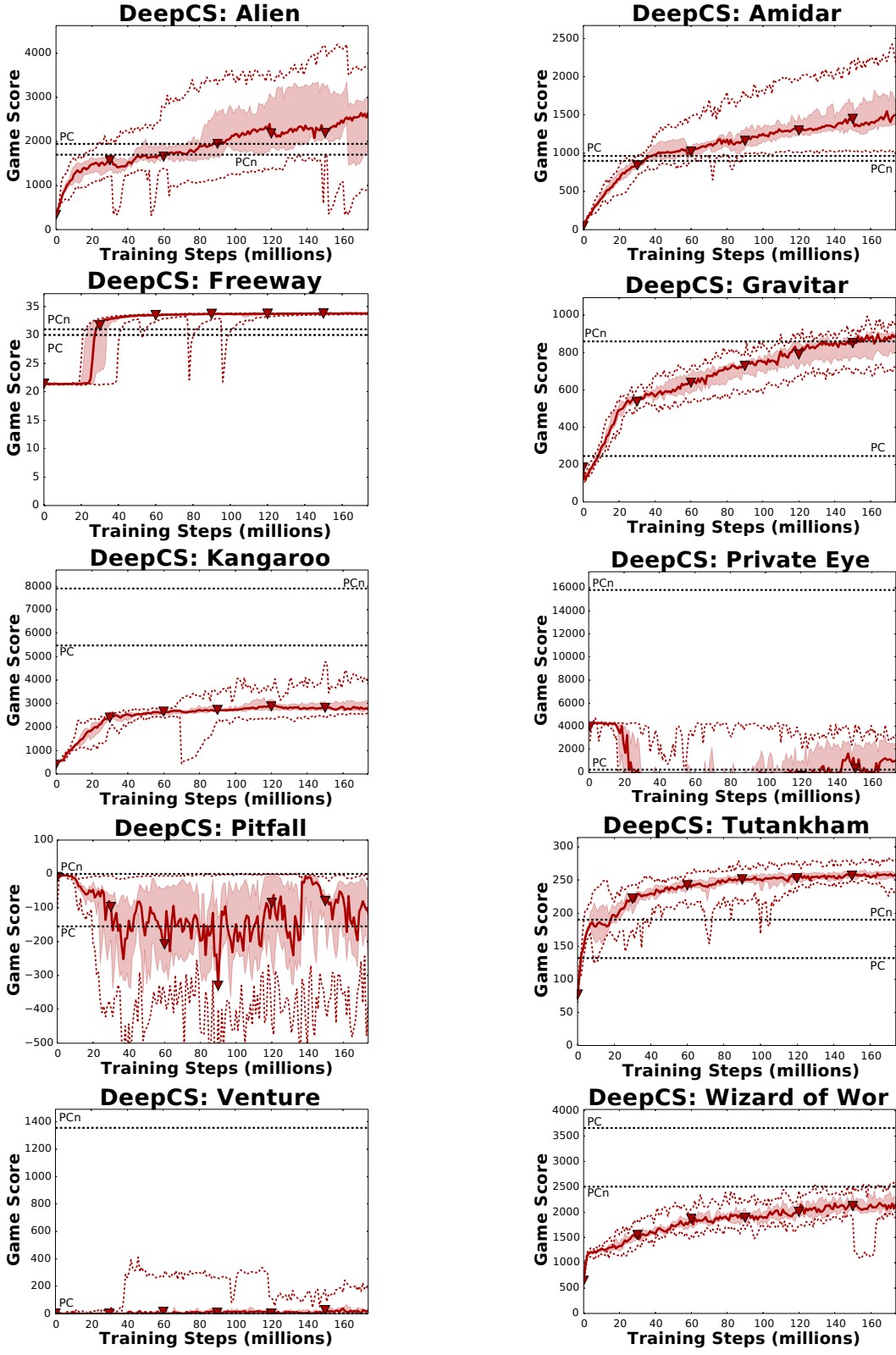

Supporting Figure S1: **DeepCS performs well on Amidar, Freeway, Gravitar, and Tutankham.**
Comparisons are shown against two state-of-the-art exploration techniques (Table 1, PC and PCn).

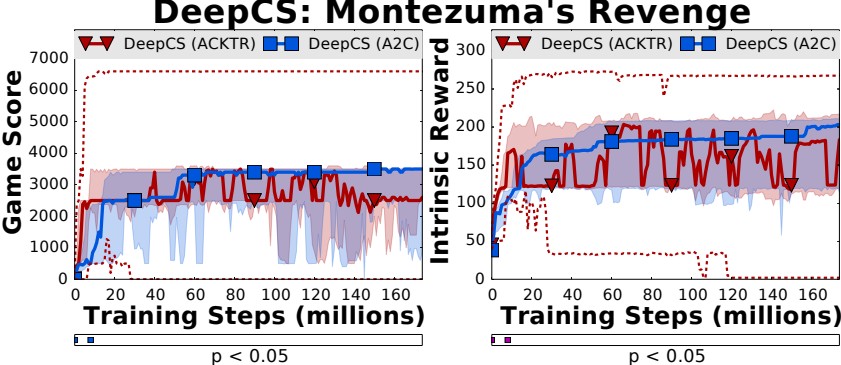

Supporting Figure S2: **DeepCS with an ACKTR implementation achieves a consistent maximum of 6600 points on Montezuma's Revenge, matching state-of-the-art maxima (Bellemare et al. (2016); Ostrovski et al. (2017)).** While ACKTR appears to be less stable overall than DeepCS with an A2C implementation, there is no statistical difference between these treatments except at the very beginning of training.

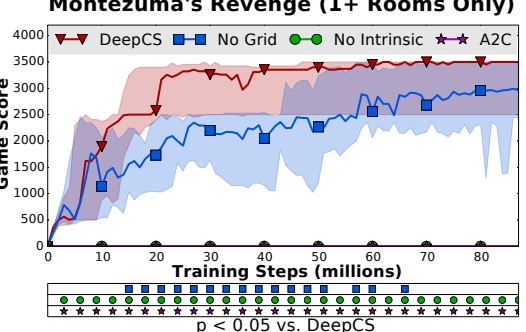

Supporting Figure S3: **Removing the curiosity grid from DeepCS in Montezuma's Revenge results in fewer agents that exit the first room.** To generate this plot, all runs that did *not* exit the first room were removed; No Intrinsic and A2C were retained for visual reference. DeepCS and No Grid are not significantly different after 70 million training steps, indicating that when agents *do* exit the first room, they eventually perform just as well as DeepCS—even without a curiosity grid input.

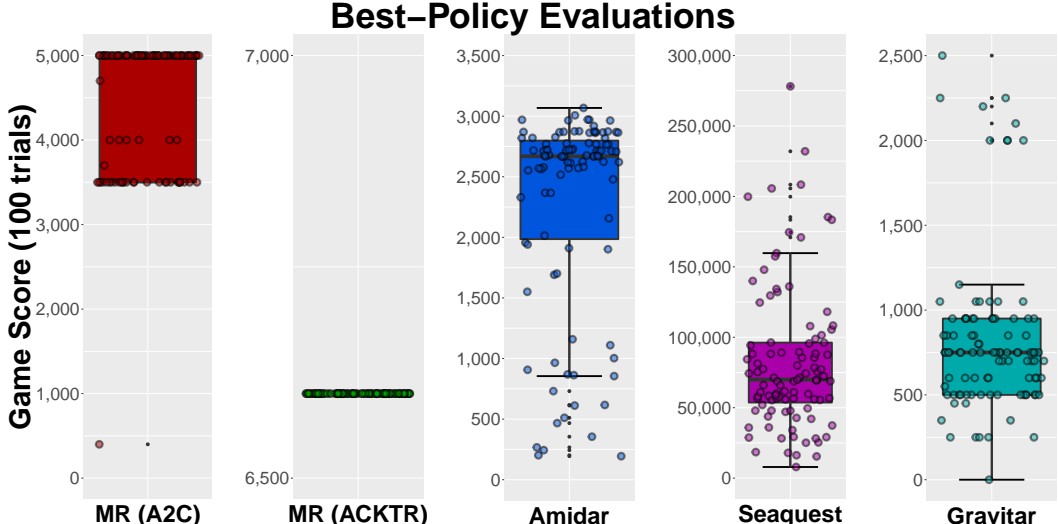

Supporting Figure S4: **A2C/ACKTR produce stochastic policies; in DeepCS, these policies sometimes perform much better than would be expected from training scores.** For each algorithm, the best single policy across all runs was evaluated 100 times with different random seeds; box plots indicate the distribution of game scores and colored dots represent data points (jittered for clarity).

