# OpenReview forum: "Deep Curiosity Search: Intra-Life Exploration Can Improve Performance on Challenging Deep Reinforcement Learning Problems"
_ICLR.cc/2019/Conference_

### Official Review · AnonReviewer3 · 2018-11-02
**More experiments might be required.**

**Rating:** 5
**Confidence:** 1

**Review:**

The reinforcement learning tasks with sparse rewards are very important and challenging. The main idea of this work is to encourage intra-life novelty. The authors introduce the curiosity grid and the intrinsic reward term so that the agent can explore toward unvisited states at every episode.

However, the results are not enough to be accepted to ICLR having a very high standard. In Section 3, the authors compare the game scores of DeepCS proposed in this paper only against to A2C. There are some RL algorithms reported to be better than A2C. For instance, I would like to see the comparison between DeepCS and SmartHash by Tang et al 2017.

=================================================================================================
I've read the rebuttal. I updated my score but still not vote for accept.

This paper is not my main research area. Very unfortunately, this paper was assigned to me. The main issue of this paper is the fair comparisons with other works. However, I don't have enough knowledges to judge this point.  So please assess this paper with other reviewers comments.

---

> ### Author Response · Authors · 2018-12-02
> **Clarifications of the algorithm and methodology**
>
> Due to the overlap between reviewer comments, we decided to address all concerns in a single response (please see above).

---

### Official Review · AnonReviewer1 · 2018-11-02
**Interesting approach to improve exploration in RL with unfair advantage**

**Rating:** 5
**Confidence:** 3

**Review:**

Summary:
The authors look at the problem of exploration in deep RL. They propose a “curiosity grid” which is a virtual grid laid out on top of the current level/area that an Atari agent is in. Once an agent enters a new cell of the grid, it obtains a small reward, encouraging the agent to explore all parts of the game. The grid is reset (meaning new rewards can be obtained) after every roll out (meaning the Atari agent has used up all its lives and the game restarts).

The authors argue that this method enables better exploration and they obtain an impressive score on Montezuma’s Revenge (MR).

Review:
The paper contains an extensive introduction with many references to prior work, and a sensible lead up to the introduced algorithm. The algorithm itself seems to work well and some of the results are convincing. I am a bit worried about the fact that the agents have access to their history of locations (“the grid”). The authors mention that none of the methods they compare against has this advantage and it seems that in a game that rewards exploration directly (MR) this is a large advantage.

The authors comment on this advantage in section 3 and found that removing intrinsic rewards hurt performance significantly. Only removing the grid access made results on MR very unstable. However in order to compute the intrinsic rewards, it still seems necessary to access the location of the agent, meaning that implicitly the advantage of the method is still there.

I was wondering if the authors find that the agents are forcibly exploring the entire environment during each rollout? Even if the agent knows what/where the actual goal is. There is a hint to this behaviour in section 4, on exploration in sparse domains.

The future work section mentions some interesting improvements, where the agent position is learned from data. That seems like a promising direction that would generalise beyond Atari games and avoids the advantage.

Nits/writing feedback:
- There is no need for such repetitive citing (esp paragraph 2 on page 2). Sometimes the same paper is cited 4 times within a few lines. While it’s great that so much prior work was acknowledged, mentioning a paper once per paragraph is (usually) sufficient and increases readability.
- I think the comparison between prior lifetimes and humans mastering a language doesn’t hold up and is distracting

####
Revision:

The rebuttal does little to clarify open questions:
1. Both reviewer 2 and I commented on the ablation study regarding the grid but received no reply.
2. I am not convinced this method is sufficiently new, given that there are other methods that try to directly reward visiting new states.
3. The authors argue in their rebuttal that "the grid" is a novel idea that warrants investigation, but remark in figure 5 that likely it isn't the key aspect of their algorithm. This seems contradictory.

---

> ### Author Response · Authors · 2018-12-02
> **Clarifications of the algorithm and methodology**
>
> Due to the overlap between reviewer comments, we decided to address all concerns in a single response (please see above).

---

### Official Review · AnonReviewer2 · 2018-11-05
**Interesting idea, weak experimental evaluation**

**Rating:** 5
**Confidence:** 3

**Review:**

This paper proposes use of intra-life coverage (an agent must visit all locations within each episode) for effective exploration in Atari games. This is in contrast of approaches that use inter-life coverage or curiosity metrics to incentivize exploration. The paper shows detailed results and analysis on 2 Atari games: Montezuma’s Revenge and Seaquest, and reports results on other games as well.

Strengths
1. Intuitively, the idea of intra-life curiosity is reasonable. The paper pursues this idea and provides experimental evidence towards it on 2 Atari games. It is able to show compelling improvements on the challenging Montezuma’s Revenge game.

Weaknesses
1. The two primary comparison points are missing:
1a. Comparison to other exploration methods. A number of methods that use state visitation counts (also referred to as diversity, eg. [A,B]), or prediction error (also referred to as curiosity, eg [C]) have been proposed in recent years. It is important to place the contributions in this paper in context of these other works. A number of these references are missing and no experimental comparison to these methods has been made.

1b. Comparison between inter and intra life curiosity. One of the central motivation is the utility of intra-life curiosity vs inter-life curiosity, yet no comparisons to this effect have been provided.

2. Additionally, the paper employs a custom way of computing coverage (or diversity). It is in terms of location of agent on the screen, as opposed to featurization of the full game screen as used in prior works. It is possible that a large part of the gain comes from the clever design of the space for computing intrinsic exploration reward. The paper tries to control for it, however that description is rather short and vague (not clear how the proposed reward is computed without there being a grid, or how is the grid useful without the intrinsic reward). More details should be provided, and when comparisons to past works or inter-life curiosity are made, this should be controlled for. The two ideas (use of grids, and intra-life curiosity vs inter-life curiosity) should be independently investigated and put in context of past work.

3. I will encourage investigation on a more varied set of tasks. Perhaps, also using some MuJoCo environments, or 3D navigation environments. Table 1 tries to provide some comparisons on Atari, however number of samples is different for different methods making the comparisons invalid. Additionally, all of these are still on Atari.

[A] Diversity is All You Need: Learning Skills without a Reward Function Benjamin Eysenbach, Abhishek Gupta, Julian Ibarz, Sergey Levine

[B] EX2: Exploration with Exemplar Models for Deep Reinforcement Learning Justin Fu, John D. Co-Reyes, Sergey Levine

[C] Curiosity-driven Exploration by Self-supervised Prediction Deepak Pathak, Pulkit Agrawal, Alexei A. Efros and Trevor Darrell International Conference on Machine Learning (ICML), 2017

---

> ### Author Response · Authors · 2018-12-02
> **Clarifications of the algorithm and methodology**
>
> Due to the overlap between reviewer comments, we decided to address all concerns in a single response (please see above).

---

### Author Response · Authors · 2018-11-26
**Clarifications of the algorithm and methodology:**

Thank you for your insights on our manuscript. Overall, we believe this paper takes an important and necessary first step by showing that the concept of Curiosity Search (rewarding agents for going everywhere within their lifetime) can improve performance on human-level problems. This was not a given; the pressure to visit new places might instead have caused agents e.g. to explore dead-ends at the expense of game rewards. Thus, the emphasis is not that intra-life exploration is superior to other directed exploration algorithms, but instead that this approach is an interesting new alternative for producing high-quality policies in sparse reward domains. With this foundation in place, we believe that future work can improve the generality of Curiosity Search (e.g. by learning positional information from an auto-encoder) and determine under which circumstances it can perform better than other directed exploration techniques.

Below, we address specific issues:

We did perform a comparison of intra-/inter-life novelty (data and details are in Supporting Information section S5; we will add a summary of those results to the main text in a future draft). Specifically, we examined never resetting the grid to simulate across-training novelty. In this setup, the algorithm obtained no rewards in Montezuma's Revenge. We believe the exploration rewards near the start were simply consumed too early and then failed to help throughout the rest of training.

We compared Curiosity Search against A2C (rather than other exploration algorithms) because a high-quality and well-tuned version of A2C was already available in OpenAI Baselines, allowing us to focus on implementing DeepCS rather than debugging/tuning the underlying algorithms (which can take considerable time in deep reinforcement learning). Along these lines, we found techniques like the pseudo-counts of Bellemare et al. too difficult/costly to implement due to their complexity and we also would have had to search for several unreported hyperparameters. The tabular comparisons against other algorithms in the manuscript serve only as a general indicator of how DeepCS fits in with other directed and undirected exploration algorithms. The emphasis is not that Curiosity Search is superior or inferior to those algorithms, but rather that it is an interesting new idea offering competitive performance on human-level problems and is worthy of further study.

The reason we add a curiosity grid (the visual input telling agents where they have not yet been), as opposed to adopting existing techniques in the field, is twofold. First, this component is an Atari analogue of the intra-life novelty compass presented in the original Curiosity Search work. Since this paper examines whether Curiosity Search can scale to human-level problems in deep RL, it makes sense that all components of that work should be represented—including the curiosity grid. Second, we believe that adding a visual memory of where agents have been is an interesting idea in its own right, one that could improve performance even when applied to other directed exploration techniques provided we can generate the positional information directly from pixels. We agree with the first reviewer that this use of a grid merits additional, more detailed investigation in future work.

---

### Meta-Review · Area_Chair1 · 2018-12-14

**Confidence:** 4
**Recommendation:** Reject

**Metareview:**

Pros:
- novel idea of intra-life curiosity that encourages diverse behavior within each episode rather than across episodes.

Cons:
- privileged/ad-hoc information (RAM state, distinguishing rooms)
- lack of sufficient ablations/analysis
- insufficient revision/rebuttal

The reviewers reached consensus that the paper should be rejected in its current form.